# Muscle Activation Differences Between CKCUEST and Modified CKCUEST: A Pilot Study

**DOI:** 10.3390/healthcare13080922

**Published:** 2025-04-17

**Authors:** Samuel Eloy Gutiérrez-Torre, Miguel Ángel Lozano-Melero, Maria Gómez-Jiménez, Daniel Manoso-Hernando

**Affiliations:** 1Department of Physiotherapy, Centro Superior de Estudios Universitarios La Salle, Universidad Autónoma de Madrid, 28023 Madrid, Spain; 201008665@campuslasalle.es (S.E.G.-T.); maria.gomez.jimenez@lasallecampus.es (M.G.-J.); daniel.m@lasallecampus.es (D.M.-H.); 2CranioSPain Research Group, Centro Superior de Estudios Universitarios La Salle, Universidad Autónoma de Madrid, 28023 Madrid, Spain

**Keywords:** CKCUEST, EMG, shoulder

## Abstract

Background/Objectives: The validity of shoulder orthopaedic tests to establish a diagnosis has recently been challenged. For this reason, functional tests, such as the Closed Kinetic Chain Upper Extremity Stability Test (CKCUEST), have started to be used in clinical settings. The aim of this study is to compare the electromyography (EMG) activity during the CKCUEST and the modified CKCUEST in a healthy adult population. Methods: Ten male (age: 26.6 ± 4.8) and ten female participants (age: 24.2 ± 6.0) were recruited from a university setting. The Edinburgh Handedness Inventory, the International Physical Activity Questionnaire and the percentage of activation of the maximum voluntary contraction of the infraspinatus, anterior deltoid and upper trapezius, of both upper limbs, throughout the CKCUEST and modified CKCUEST were analysed. Results: The percentage of activation of the infraspinatus (*p* < 0.01), anterior deltoid (*p* < 0.01) and upper trapezius (*p* < 0.01) in both sides was significantly higher in the CKCUEST compared to the modified CKCUEST. No differences were observed between laterality and the activation percentage of the infraspinatus (*p* > 0.05), anterior deltoid (*p* > 0.05) and upper trapezius (*p* > 0.05) in both sides during the CKCUEST. Conclusions: The results of this research showed a higher percentage of EMG activation during the CKCUEST compared to the modified CKCUEST in all the muscular structures analysed, regardless of the participants’ hemibody.

## 1. Introduction

The shoulder represents the third most prevalent region of musculoskeletal disorders, with an estimated 20% of the population reporting shoulder pain at some stage in their lifetime [1]. Of these cases, 20% to 40% are attributable to pathologies involving the rotator cuff complex, a condition known to cause substantial functional impairment and reduced quality of life [2,3].

Historically, rotator cuff diagnosis has been based on physical examinations and imaging. Nevertheless, exclusive reliance on imaging for diagnostic purposes presents significant challenges, as structural abnormalities in the rotator cuff are frequently observed even in asymptomatic individuals [4,5].

In regards to physical examinations of the shoulder, about 180 orthopaedic shoulder tests have been described in the literature [6], which leads to great confusion when naming and referring to the same test [7], even when making diagnostic clusters [8]. However, the validity of these tests based on strong methodological designs, such as meta-analyses, are scarce, and therefore, they should be considered as symptom provocation tools rather than diagnostic tests [3,9,10,11].

For this reason, functional tests have begun to be used [6,12], such as the Closed Kinetic Chain Upper Extremity Stability Test (CKCUEST) (ICC = 0.82–0.98) [13], the Upper Limb Rotation Test (ULRT) (ICC = 0.90–0.99) [14], Kerlan-Jobe Orthopaedic Clinic Shoulder and Elbow Score (KJOC) (ICC = 0.50–0.93) [15], the Y-Balance Test Upper Quarter (YBT-QT) (ICC = 0.91–0.99) [16], and the Seated Medicine Ball Throw (SMBT) (ICC = 0.88–0.99) [17].

The CKCUEST is a functional and dynamic test used for the assessment of the shoulder which usually assesses athletes whose sporting activity mainly involves the use of this upper limb [18]. This test consists of placing the participants in a push-up starting position, from which they must touch one hand with the opposite hand and then perform the same procedure with the contralateral side [18] for fifteen seconds [13].

Previous studies have shown that this test can serve as a tool for shoulder assessments of patients with pain, as well as for follow-up as part of a care routine or even rehabilitation process, achieving higher movement efficiency values of the upper extremity [19,20].

On the other hand, electromyography (EMG) studies have shown how the shoulder girdle muscles are activated throughout shoulder movements [21]. In the same direction, Kinsella et al. found lower muscle EMG activity in patients with painful shoulder syndrome [22]. Similarly, other studies have observed differences in upper trapezius, infraspinatus and serratus anterior muscle activation in patients with shoulder pathology versus asymptomatic subjects as measured by EMG in a variety of shoulder functional movements [23,24].

However, although the importance of assessing this musculature is evident in the scientific literature [22,23,24] and functional tests such as the CKCUEST have been proposed to assess shoulder pathology [25,26], to the authors’ knowledge, there are no studies that assess EMG activity in the CKCUEST in healthy adults. The rationale for analysing EMG data during the CKCUEST is based on its ability to provide an objective assessment of muscle activation patterns, which are crucial for understanding neuromuscular demands during closed kinetic chain tasks.

Therefore, the aim of this study is to compare the percentage of activation of the maximum voluntary isometric contraction (MVIC) of the infraspinatus, anterior deltoid and upper trapezius muscles during the execution of the CKCUEST and the modified CKCUEST in a healthy adult population.

## 2. Materials and Methods

### 2.1. Participants

To carry out this observational study, we followed the STROBE (Strengthening the Reporting of Observational Studies in Epidemiology) guidelines [27], which are a series of recommendations for writing up studies to ensure that the necessary information is provided on how the study was conducted, what was found and what was not found [27]. The required sample size was estimated based on previous recommendations for pilot studies [28].

For this pilot study, a total of 20 healthy subjects (10 male and 10 female participants) were recruited through non-probabilistic consecutive sampling stratified by sex of the student population of the Centro Superior de Estudios Universitarios de la Salle (CSEULS), where the laterality of the healthy adult participants was determined using the Edinburgh Handedness Inventory (EHI) manual dominance questionnaire [29], adapted to Spanish [30].

As inclusion criteria, participants were accepted if they presented the following:-A good state of health;-An asymptomatic state in the region of the shoulder complex (both right and left);-Age of 18 years or more.

As exclusion criteria, participants were rejected if they presented the following:-Shoulder pathology (rotator cuff-related shoulder pain, frozen shoulder, severe shoulder osteoarthritis, cervical radiculopathy, shoulder instability, upper limb neuropathy or acromio-clavicular joint pathology);-Sensory and/or motor deficits;-Age of over 65 years;-Shoulder surgery less than 6 months ago;-Previous steroid injections in the last 3 months;-Diagnoses related to the cervical spine or upper limbs;-Pregnancy;-Comorbidities such as arthritis, rheumatoid arthritis and/or fibromyalgia;-Systemic diseases such as diabetes mellitus and/or thyroid diseases;-Dementia or severe psychiatric illness and any other illness that could interfere with their understanding of and/or participation in the study;-Refusal to sign informed consent.

### 2.2. Measuring Instruments

#### 2.2.1. Edinburgh Handedness Inventory (EHI)

The Spanish-validated version of the EHI consists of a 10-item self-report questionnaire on manual dominance or preference of one hand over the other in manual activities [30].

This questionnaire measures the preference of the use of one hand over the other in different activities.

The original EHI asks the participant to mark each item in one of two columns with the symbol + depending on whether it refers to the right or left hand. If the patient does not use either of their hands, they are asked to mark with ++ the hand for which they have a preference. If the patient is indifferent, they are asked to mark + in both columns.

It has been shown that these instructions can lead to misunderstandings; therefore, in the Spanish edition, it has been changed to a 5-point scale format as follows (1 = always right, 2 = usually right, 3 = both equally, 4 = usually left and 5 = always left). The laterality quotient (LQ) is estimated on the basis of the following formula:*LQ* = (*R* − *L*)/(*R* + *L*) × 100

The *LQ* classifies the laterality of each participant’s hand into left-handed (−100 to −61), ambidextrous (−60 to 60) or right-handed (61 to 100) [30].

#### 2.2.2. International Physical Activity Questionnaire

The International Physical Activity Questionnaire (IPAQ) is a self-reported questionnaire for the measurement of activity in subjects aged 15–69 years, with a long version with 31 items and a short version with 7 items, validated in Spanish [31]. This questionnaire assesses the frequency, duration and intensity of physical activity. This questionnaire provides the results of physical activity in metabolic equivalents per minute per week (METs/min/week); in the long version, the total score of each activity is divided into different domains (leisure-related, household and gardening, occupational and transport activities). Both categorise physical activity as low, moderate and high, providing the number of MET minutes/week and thus indicating the amount of physical activity [32].

#### 2.2.3. Electromyography

Surface electromyographic data were collected with a BTS Bioengineering^®^ FreeEMG 300 system (BTS Bioengineering, Milan, Italy). All electrodes were placed over the muscle bellies, in line with the orientation of muscle fibres, as previously described in the literature [33].

For signal processing, it was amplified (gain = 1000), rectified and then band-pass-filtered (Butterworth; 20–400 Hz). The envelope of the whole EMGs signal was calculated with a fixed (for the MVIC test) and dynamic (for the CKCUEST test) 500 ms time window. Finally, the normalised root mean square (%RMS) of the 5 central seconds of each CKCUEST stage was calculated and expressed as a percentage (%) of the corresponding MVIC test (%RMS).

### 2.3. Procedure

Following the EHI [30] and IPAQ questionnaires [32], a standardised warm-up consisting of multiplane shoulder movements supervised by the researchers was performed.

Next, prior to the CKCUEST and modified CKCUEST, a test was performed to calculate the MVIC for each muscle using manual resistance exerted by the examiner.

Participants performed three MVICs per muscle, and all muscles were tested randomly [34]. EMG activity was measured for 5 s, taking as reference the central 3 s and leaving 1 min of rest between repetitions [35,36].

For each muscle, the highest level of activity generated during the 3 MVIC assessment positions was used for normalisation [37]. The same researcher was responsible for all MCVI measurements to ensure test consistency.

The MVIC calculation test for each muscle was performed according to Cools et al. (2020) [33]:Anterior Deltoid: Patient is seated, both of their feet are supported, elbow is at 90° flexion and shoulder is in neutral position. Resistance is applied proximal to the elbow in the posterior direction (resisting shoulder flexion);Infraspinatus: Patient is in seated position, both of their feet are supported, the elbow is at 90° of flexion and the shoulder is in neutral position. Resistance is applied proximal to the wrist in medial direction (resisting external rotation);Upper trapezius: Patient is seated, both feet are supported, the elbow is extended and the shoulder is at 90° abduction. Resistance is applied proximal to the elbow in the caudal direction (resisting abduction).

Pictures detailing the electrodes’ position are presented in Figure 1.

The CKCUEST was performed in a push-up position, keeping the back straight parallel to the floor, hands spaced 91.44 cm (36 inches) apart and both upper limbs perpendicular to the floor and over the hands [26]. The exercise started with 5 s rest, then 3 s with the left hand touching the right hand, 3 s rest and 3 s with the right hand touching the left hand, controlled by a metronome at 60 Hertz. In total, 5 repetitions were performed, the first and last of which were eliminated from the analysis [33,38].

The mean for the EMG peak activity of each muscle during the closed kinetic chain phase was obtained for the three consecutive repetitions. The mean obtained was then expressed as a percentage of the MVIC.

The modified CKCUEST was performed in a push-up position with knees flat on the floor, keeping the back straight and parallel to the floor, hands spaced 91.44 cm (36 inches) apart, and both upper limbs perpendicular to the floor and over the hands [26,39]. The task consists of the same phases as the CKUEST, which were also controlled by a metronome with 5 repetitions, of which the first and the last were eliminated in the same way [33,38].

All participants performed the CKCUEST and the modified CKCUEST, which were randomly ordered by flipping a coin; there was a two-minute rest between the performance of each test.

Similarly to the CKCUEST analysis, the mean for the EMG peak activity of each muscle during the closed kinetic chain phase was obtained for the three consecutive repetitions throughout the modified CKCUEST. The mean obtained was then expressed as a percentage of the MVIC.

### 2.4. Statistical Analysis

Statistical Package of Social Sciences (SPSS) software, version 24.0 (IBM, Armonk, NY, USA, was used for statistical analysis. A *p*-value < 0.05 for a 95% confidence interval (CI) was determined as statistically significant, using an alpha-type error of 0.05.

The Shapiro–Wilk test was used to determine the normality distribution of the data values obtained. Data with a parametric distribution were represented by mean, standard deviation and upper and lower limits of the 95% CI (and minimum–maximum range). As statistical tests for the parametric data, Student’s *t*-test for related samples was used to perform the comparison analysis between the CKCUEST test and the modified CKCUEST test, and Student’s *t*-test for independent samples was used to compare sex, as well as to compare the percentage of activation of the infraspinatus, anterior deltoid and upper trapezius muscles according to the laterality of the participants in the present study.

On the other hand, for the description of the non-parametric data, we used the median, the interquartile range and the upper and lower limits of the 95% (CI) (and minimum–maximum range). For their analysis, we used the Wilcoxon test for related samples for the analysis of comparison between the CKCUEST test and the modified CKCUEST test and the Wilcoxon Mann–Whitney U test for the analysis of sex, as well as to compare the percentage of activation of the infraspinatus, anterior deltoid and upper trapezius muscles as a function of the laterality of the participants in the present study.

## 3. Results

A total of 20 participants were recruited. Ten men and ten women met our inclusion criteria. A description of the participants can be found in Table 1. The variables weight, height, body max index (BMI) and IPAQ showed a normal distribution; hence, a Student’s *t*-test for independent samples was used to assess the differences between the two groups. On the other hand, the EHI and age did not follow a normal distribution; therefore, the Wilcoxon Mann–Whitney U test was employed to assess the differences between the two groups.

The percentage of activation of the infraspinatus, anterior deltoid and upper trapezius on both sides was significantly higher in the CKCUEST compared to the modified CKCUEST (*p* < 0.01), as shown in Table 2.

Comparisons between laterality and the percentage of activation of the infraspinatus, anterior deltoid and upper trapezius muscles in the CKCUEST test are shown in Table 3.

Comparisons between sex and the percentage of activation of the infraspinatus, anterior deltoid and upper trapezius muscles in the CKCUEST test are shown in Table 4.

## 4. Discussion

This study has identified significant muscle activation percentage differences in the infraspinatus, anterior deltoid and upper trapezius muscles during the performance of the CKCUEST and the modified CKCUEST in healthy participants. No statistical differences in the percentage of activation in the muscles studied were shown between men and women or between right-handed and ambidextrous subjects. Equally, no statistical differences in IPAQ scores were found between men and women. Given that physical activity can influence neuromuscular responses, using the IPAQ allowed us to categorize and control for activity levels, ensuring that any differences observed in muscle activation were due to the studied variables rather than variations in habitual physical activity [40].

According to our results, the EMG activity of the infraspinatus, anterior deltoid and upper trapezius muscles bilaterally observed during the performance of the CKCUEST is higher than the activation percentages obtained during the performance of the modified CKCUEST in the same muscle regions in a healthy adult population. This could be explained by the fact that, throughout the CKCUEST, the push-up position demands greater muscle activity to stabilise the shoulder complex than during the modified CKCUEST. In the latter, the participants were quadrupedal with their knees flat on the floor; hence, the muscle activity to stabilise the shoulder is lower [41].

To the authors’ knowledge, this is the first study to compare the EMG activity of the shoulder during the performance of such dynamic tests in a healthy adult population.

Shoulder pain assessments based on shoulder range of motion, strength, mapping pain and shoulder functional tests have been proposed in the literature [25,42,43], and statistical differences have been found in recent studies [44]. Hence, the CKCUEST seems to be an ideal tool for functional shoulder assessment. According to our results, the modified CKCUEST might be more optimal for shoulder pain patients as a decreased EMG activity in all muscles studied was shown compared to the CKCUEST, as has been proposed by Tucci et al. (2014) [26]. Exercise progression to the CKCUEST seems logical to increase the loading capacity throughout the conservative treatment, as loading progression has been proposed as a key factor in shoulder pain management [45,46]. Moreover, an association between the CKCUEST and shoulder strength has been established [47]. Also, a moderate correlation has been observed between the ULRT and CKCUEST scores (r range = 0.505–0.589), as it has been shown that CKCUEST can account for 30.6% to 37.8% of the variance in the ULRT performances. The ULRT was designed to promote weight bearing, requiring shoulder motor control and stability and involving the entire kinetic chain in a more complex shoulder position at 90° abduction and 90° external rotation [14].

The CKCUEST should be considered as an assessment tool in patients whose pain severity and irritability allow them to perform the functional test as well as in healthy subjects [26]. However, as only healthy participants were included in the study, further studies are needed to evaluate both the CKCUEST and the modified CKCUEST in shoulder pain patients.

In line with our findings, Gorman et al. (2012) found no differences in EMG activity between sex and hand dominance during the performance of the UQY-BT [16]. However, Tucci et al. (2014) found that healthy females performed a higher number of touches throughout the CKCUEST than healthy males [26], possibly due to anthropometric variables such as the wingspan differences between sex. Hence, even though healthy women seem to perform a greater number of touches than men during the CKCUEST, the EMG activation in the infraspinatus, upper trapezius and anterior deltoid does not appear to be different between groups.

Our results have implications for clinical practice and future studies. The CKCUEST has been proposed as a functional test for the physical examination of patients suffering shoulder pain [26] and to evaluate shoulder performance in athletes [48]; however, subjects with low physical activity levels and patients with shoulder dysfunction might find the CKCUEST difficult to properly perform [26]. Our findings provide evidence that the modified CKCUEST induces significantly lower EMG activity in the infraspinatus, anterior deltoid and upper trapezius muscles compared to the CKCUEST. This suggests that the modified CKCUEST is a highly suitable option for physical functional assessment in individuals with shoulder pain or low physical activity levels. Furthermore, structured progression from the modified CKCUEST to the CKCUEST should be considered in shoulder rehabilitation to optimize tissue loading capacities, as this test has demonstrated a strong correlation with shoulder strength [47].

Therefore, future investigations should evaluate the difference in EMG activity between patients with shoulder pain and those who are asymptomatic, as well as determine whether the inclusion of CKCUEST progression within a shoulder rehabilitation process improves variables such as pain and function in patients with shoulder pain.

### Limitations

At present, this research is a pilot study, in which the sample size calculated for the main variable has not been reached. Furthermore, while the modified CKCUEST may be suitable for patients with shoulder pain, the study only included healthy participants. For these reasons, the results of this research should be taken with caution, reducing the external validity of the data and, therefore, limiting its interpretation.

Another limitation to be taken into account is the design of the tests themselves, which does not take into account the size of the upper limbs of the participants, which could influence changes in the activation of the regions analysed or other future variables to be measured. Therefore, it would be interesting to evaluate muscle activity, taking into account the participants’ wingspan, as well as the design of the functional test itself. One of the primary limitations of this study is its cross-sectional observational design, which inherently restricts the ability to establish causal relationships. Additionally, the absence of longitudinal follow-up and the lack of complementary methodological approaches limit the generalizability and applicability of the findings. Future research should consider incorporating alternative designs that allow for interventional assessments and long-term monitoring. In particular, it would be valuable to evaluate muscle activity during the execution of these dynamic tests using more advanced methodologies, both evaluative and interventional, to gain deeper insights into neuromuscular function and adaptation.

The CKCUEST consists of assessing the maximum number of touches throughout fifteen seconds in a determined position; however, due to the limitations of EMG, to accurately pin down the precise peak muscle activity during the test, we proceeded to analyse the data as previously proposed in the literature as follows: starting with 5 s of rest, then 3 s with the left hand touching the right hand, 3 s of rest and 3 s with the right hand touching the left hand, controlled by a metronome at 60 Hertz. In total, five repetitions were performed, the first and last of which were eliminated in the analysis, and the mean for the EMG peak activity of each muscle was obtained during the closed kinetic chain phase [33,38].

Finally, given the nature of the pilot study and the type of sampling stratified by sex, it was not possible to collect participants, as verified by the EHI questionnaire, with upper limb dominance characteristics that would define them as left-handed. This has compromised the analysis of these variables and, thus, one of the secondary objectives of the study. Therefore, it would be interesting to carry out stratified sampling by type of upper limb dominance in order to be able to draw more solid conclusions.

## 5. Conclusions

The results of this research showed a higher percentage of EMG activation during the CKCUEST compared to the modified CKCUEST in all the muscular structures analysed, regardless of the participants’ hemibody.

## Figures and Tables

**Figure 1 healthcare-13-00922-f001:**
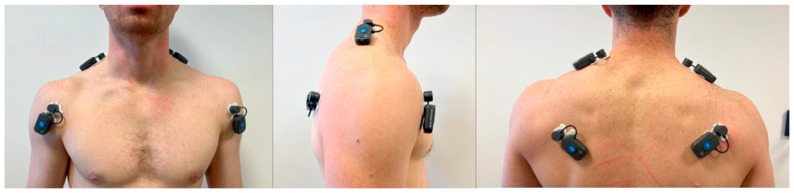
Electrode placement for EMG recording.

**Table 1 healthcare-13-00922-t001:** Socio-demographic variables, *EHD* and *IPAQ*.

	Men (*n* = 10)	Women (*n* = 10)	*p*-Value
EHD (r, l, a)	18.6 ± 8.6 ^a^	13.5 ± 11.0 ^c^	0.462 ^d^
Age (years)	26.6 ± 4.8 ^a^	24.2 ± 6.0 ^c^	0.250 ^d^
Weight (kg)	74.5 ± 9.9 ^a^	57.6 ± 6.8 ^a^	0.000 ^b^
Height (cm)	179.9 ± 7.1 ^a^	165.4 ± 6.0 ^a^	0.000 ^b^
BMI (kg/m^2^)	22.9 ± 1.7 ^a^	21.0 ± 2.1 ^a^	0.049 ^b^
*IPAQ* (METs/min/week)	3532.0 ± 1,291,606 ^a^	4160.8 ± 1637.0 ^a^	0.353 ^b^

Abbreviations: EHD, Edinburgh Handedness Inventory (measured as r, right-handed; l, left-handed; a, ambidextrous); BMI, body mass index; IPAQ, International Physical Activity Questionnaire. ^a^ Data with parametric distribution are represented by mean, standard deviation and 95% confidence interval. ^b^ Student’s *t*-test for independent simples. ^c^ Data with nonparametric distribution are represented by median, interquartile range and 95% confidence interval. ^d^ Wilcoxon Mann–Whitney U-test.

**Table 2 healthcare-13-00922-t002:** Comparisons between the percentage of activation of the infraspinatus, anterior deltoid and upper trapezius muscles between the CKCUEST and modified CKCUEST.

	Activation % CKCUEST (*N* = 20)	Activation % Modified CKCUEST (*N* = 20)	*p*-Value
Right hemibody			
- Infraspinatus	39.3 ± 15.0 ^a^ (32.3−46.4)	15.7 ± 20.2 ^b^ (15.9−24.3)	0.000 ^†^
- Anterior deltoid	61.9 ± 24.4 ^a^ (52.4−71.5)	24.7 ± 15.7 ^a^ (17.3−32.1)	0.000 *
- Upper trapezius	11.1 ± 15.6 ^b^ (10.8−25.6)	6.3 ± 7.1 ^b^ (5.6−11.9)	0.000 ^†^
Left hemibody			
- Infraspinatus	38.5 ± 16.6 ^a^ (30.7−46.3)	15.3 ± 8.3 ^b^ (11.3−25.8)	0.000 ^†^
- Anterior deltoid	62.8 ± 21.5 ^a^ (52.7−72.9)	28.8 ± 19.6 ^a^ (19.6−38.0)	0.000 *
- Upper trapezius	15.3 ± 22.7 ^b^ (14.6−37.5)	10.6 ± 16.8 ^b^ (7.5−24.9)	0.000 ^†^

Abbreviations: CKCUEST, Closed Kinetic Chain Upper Extremity Stability Test. ^a^: Data with parametric distribution are represented by mean, standard deviation and 95% confidence interval. ^b^: Data with nonparametric distribution are represented by median interquartile range and 95% confidence interval. *: Student’s *t*-test for related samples. ^†^: Wilcoxon test for related samples.

**Table 3 healthcare-13-00922-t003:** Comparison between laterality and the activation percentage of the infraspinatus, anterior deltoid and upper trapezius muscles in the CKCUEST.

Activation Percentage in CKCUEST	Right-Handed (%) (*n* = 10)	Ambidextrous (%) (*n* = 10)	Left-Handed (%) (*n* = 0)	*p*-Value
Right hemibody				
- Infraspinatus	30.6 ± 16.5 ^a^ (26.8−50.4)	40.0 ± 14.2 ^a^ (29.8−50.2)	**-**	0.844 *
- Anterior deltoid	65.5 ± 17.9 ^a^ (52.7−78.3)	50.4 ± 23.0 ^a^ (41.9−74.9)	**-**	0.453 *
- Upper trapezius	15.6 ± 28.2 ^b^ (10.2−36.7)	9.2 ± 6.9 ^b^ (5.1−20.8)	**-**	0.082 ^†^
Left hemibody				
- Infraspinatus	35.8 ± 18.2 ^b^ (27.−54.4)	35.9 ± 14.8 ^a^ (25.3−46.6)	**-**	0.623 ^†^
- Anterior deltoid	59.4 ± 36.7 ^b^ (52.9−81.1)	58.7 ± 23.5 ^a^ (41.9−75.5)	**-**	0.364 ^†^
- Upper trapezius	3.6 ± 27.5 ^a^ (14.9−54.2)	15.6 ± 9.6 ^a^ (8.7−22.5)	**-**	0.054 *

Abbreviations: CKCUEST, Closed Kinetic Chain Upper Extremity Stability Test. ^a^: Data with parametric distribution are represented by mean, standard deviation and 95% confidence interval. ^b^: Data with nonparametric distribution are represented by median interquartile range and 95% confidence interval. *: Student’s *t*-test for independent samples. ^†^: Wilcoxon Mann–Whitney U-test.

**Table 4 healthcare-13-00922-t004:** Comparison between sex and the percentage of activation of the infraspinatus, anterior deltoid and upper trapezius muscles in the CKCUEST.

M	MVIC Women (µV) (*n* = 10)	MVIC Men (µV) (*n* = 10)	*p*-Value
Right hemibody			
- Infraspinatus	295.8 ± 110.4 ^b^ (205.9−497.7)	281.9 ± 324.8 ^b^ (146.2−760.6)	0.940 **^†^**
- Anterior deltoid	457.6 ± 183.4 ^a^ (326.5−588.8)	444.1 ± 213.8 ^a^ (291.1−597.1)	0.881 *
- Upper trapezius	426.4 ± 194.3 ^a^ (287.3−565.4)	520.8 ± 280.9 ^a^ (319.8−721.7)	0.394 *
Left hemibody			
- Infraspinatus	291.4 ± 110.3 ^a^ (212.5−370.3)	379.3 ± 190.3 ^b^ (178.5−750.0)	0.257 ^†^
- Anterior deltoid	412.1 ± 136.5 ^a^ (314.4−509.8)	470.0 ± 268.3 ^a^ (278.1−661.9)	0.553 *
- Upper trapezius	389.9 ± 188.9 ^a^ (254.8−525.0)	359.2 ± 169.9 ^b^ (214.7−517.2)	0.406 ^†^

Abbreviations: µV, microvolts; MVIC: maximum voluntary isometric contraction. ^a^: Data with parametric distribution are represented as mean, standard deviation and 95% confidence interval. ^b^: Data with nonparametric distribution are represented as median interquartile range and 95% confidence interval. *: Student’s *t*-test for independent samples. ^†^: Wilcoxon Mann–Whitney U-test.

## Data Availability

The raw data supporting the conclusions of this article will be made available by the authors on request.

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
