# Peer review of "Muscle Activation Differences Between CKCUEST and Modified CKCUEST: A Pilot Study"

_healthcare, 2025, doi:10.3390/healthcare13080922_

Round 1
Reviewer 1 Report
Comments and Suggestions for Authors
Dear Authors,
I had the opportunity to review this interesting article of yours comparing CKCUEST and Modified CKCUEST in the clinical evaluation of shoulder function. Your results would seem to suggest that the Modified CKCUEST shows a lower EMG activity of some muscles in healthy subjects, which apparently would make it easier to perform for subjects affected by shoulder pain or dysfunction.
Below are some suggestions for improving some sections of the text:
TITLE
- “An EMG Analysi” I think the “s” at the end of the word “Analysi” is missing
ABSTRACT
- line 11: it is perhaps better to write “test” as “tests” (in the plural)
- The content is clear
- I suggest you replace all the “;” with normal “,”
INTRODUCTION
- lines 30-31: “The shoulder represents the third most prevalent region of musculoskeletal disorders, with an estimated 20% of the population reporting shoulder pain at some stage in their lifetime” can you provide a reference for this?
- line 44: “Mañoso Hernando 20/11/24 7:48” what is this? Maybe a typo to be deleted?
- line 66: the sentence even if it continues the previous one, indents like the top line, it should be corrected
MATERIALS AND METHODS:
- line 135: “BTS Bioengineering® FreeEMG 300 system” please provide the manufacturer’s data in brackets (e.g. (BTS, Location))
- line 159: “Pictures detailing the electrodes position are added in the annexes section as Figure 1.” But where is Figure 1? It is not found anywhere in the PDF I got... the figure should be inserted if there is a reference to it, otherwise remove this sentence
- lines 203-205: “Interventionary studies involving animals or humans, and other studies that require ethical approval, must list the authority that provided approval and the corresponding ethical approval code.” Is this section a typo of the journal's instructions? You should fill it out with the possible ethical approval of the study (which you have also correctly reported in the accessory section “Institutional Review Board Statement”) and even better move it to your paragraph “participants”
RESULTS:
- Line 207: “Ten men and 10 women” choose: use only the number or only letters to indicate the sample size (“10 and 10” or “ten and ten”)
DISCUSSION:
- in general ok
CONCLUSIONS:
- Ok
ACCESSORY SECTIONS:
- Data Availability Statement: “Data Availability Statements are available in section “MDPI Research Data Policies” at https://www.mdpi.com/ethics.” This is an indication of the journal, please provide a Data Availability Statement (Considering that, I believe, you have sent the collected data as Supplementary Material)
SUPPLEMENTARY MARIAL:
- In the excel file there is a page called “participantes filiacion” that reports the names of the participants (some only names, others with acronyms, others in full). Are you sure that this is ethical and respectful of patients’ privacy? If not, perhaps it would be better to remove this page.
I find your article quite interesting and relatively clearly written. I advise you to make the suggested changes and I hope you can publish it as soon as possible. Good luck with your work.
Author Response
Reviewer 1
Dear Authors,
I had the opportunity to review this interesting article of yours comparing CKCUEST and Modified CKCUEST in the clinical evaluation of shoulder function. Your results would seem to suggest that the Modified CKCUEST shows a lower EMG activity of some muscles in healthy subjects, which apparently would make it easier to perform for subjects affected by shoulder pain or dysfunction.
Below are some suggestions for improving some sections of the text:
TITLE
- “An EMG Analysi” I think the “s” at the end of the word “Analysi” is missing
Response: Thank you very much for your valuable correction. We have modified the title as follows and it is highlighted in yellow in the title page:
“Muscle Activation Differences Between CKCUEST and Modified CKCUEST: A Pilot Study.”
ABSTRACT
- line 11: it is perhaps better to write “test” as “tests” (in the plural)
Response: Again, thank you very much for your comment as it adds value to the manuscript. It has been modified in the manuscript and has been highlighted in yellow.
- The content is clear
Response: Thank you very much for your kind words.
- I suggest you replace all the “;” with normal “,”
Response: I do appreciate your comment and we have modified all the “;” with normal “,”. They have been highlighted in yellow in the manuscript.
INTRODUCTION
- lines 30-31: “The shoulder represents the third most prevalent region of musculoskeletal disorders, with an estimated 20% of the population reporting shoulder pain at some stage in their lifetime” can you provide a reference for this? Response: Thank you very much for your appreciation. We have added the following reference to justify the sentence, and it is highlighted in yellow in the manuscript in line 28:
“Luime, J.J.; Koes, B.W.; Hendriksen, I.J.M.; Burdorf, A.; Verhagen, A.P.; Miedema, H.S.; Verhaar, J.A.N. Prevalence and Incidence of Shoulder Pain in the General Population; a Systematic Review. Scand. J. Rheumatol. 2004, 33, 73–81, doi:10.1080/03009740310004667.”
- line 44: “Mañoso Hernando 20/11/24 7:48” what is this? Maybe a typo to be deleted?
Response: We appreciate the reviewer for identifying this issue. The phrase in question was an unintentional note that has now been removed from the manuscript. We apologize for any confusion this may have caused and thank you for your careful review.
- line 66: the sentence even if it continues the previous one, indents like the top line, it should be corrected
Response: We appreciate the reviewer’s feedback. In response, we have made the necessary corrections in the manuscript to address this point. Thank you for your valuable input.
MATERIALS AND METHODS:
- line 135: “BTS Bioengineering® FreeEMG 300 system” please provide the manufacturer’s data in brackets (e.g. (BTS, Location))
Response: We do appreciate you valuable comment and we have added the manufacturer´s data in brackets as follows. It has been highlighted in yellow in the manuscript in line 139.
“(BTS Bioengineering, Milan, Italy).”
- line 159: “Pictures detailing the electrodes position are added in the annexes section as Figure 1.” But where is Figure 1? It is not found anywhere in the PDF I got... the figure should be inserted if there is a reference to it, otherwise remove this sentence
Response: We appreciate the reviewer’s careful assessment. Figure 1 was unintentionally omitted from the manuscript. We have now included it in the main text to ensure clarity, particularly regarding electrode placement. Thank you for bringing this to our attention.
- lines 203-205: “Interventionary studies involving animals or humans, and other studies that require ethical approval, must list the authority that provided approval and the corresponding ethical approval code.” Is this section a typo of the journal's instructions? You should fill it out with the possible ethical approval of the study (which you have also correctly reported in the accessory section “Institutional Review Board Statement”) and even better move it to your paragraph “participants”
Response: We appreciate the reviewer’s insightful observation. We acknowledge that the sentence in question represents a general ethical guideline rather than study-specific content. In response to this suggestion, we have removed the sentence to enhance the clarity and relevance of the Methods section. Thank you for your careful review and constructive feedback.
RESULTS:
- Line 207: “Ten men and 10 women” choose: use only the number or only letters to indicate the sample size (“10 and 10” or “ten and ten”)
Response: Thank you very much for your valuable correction, we have modified the sentence as follows and it is highlighted in yellow in the manuscript in line 216.
“Ten men and ten women met our inclusion criteria.”
DISCUSSION:
- in general ok
Response: Thank you very much for your kind words
CONCLUSIONS:
- Ok
Response: Thank you
ACCESSORY SECTIONS:
- Data Availability Statement: “Data Availability Statements are available in section “MDPI Research Data Policies” at https://www.mdpi.com/ethics.” This is an indication of the journal, please provide a Data Availability Statement (Considering that, I believe, you have sent the collected data as Supplementary Material)
Response: Thank you for your valuable feedback. We appreciate your suggestion regarding the Data Availability Statement. In accordance with the journal's guidelines, we have now included the following statement in the revised manuscript, highlighted in yellow in line 340:
"The data supporting the findings of this study are available in the Supplementary Material.”
SUPPLEMENTARY MARIAL:
- In the excel file there is a page called “participantes filiacion” that reports the names of the participants (some only names, others with acronyms, others in full). Are you sure that this is ethical and respectful of patients’ privacy? If not, perhaps it would be better to remove this page.
Response: We appreciate your valuable correction, as it is true that page should have been removed. We have proceeded to delete that page.
I find your article quite interesting and relatively clearly written. I advise you to make the suggested changes and I hope you can publish it as soon as possible. Good luck with your work
Response: Thank you very much for your kind words, we do appreciate them.
We appreciate the reviewer's feedback and are open to further clarifications or additional analyses to strengthen the study.
Reviewer 2 Report
Comments and Suggestions for Authors
- The manuscript contains numerous short paragraphs consisting of only one or two sentences, which negatively impact readability. It is recommended to combine adjacent paragraphs that discuss similar topics to enhance the logical flow. If the content is insufficient, additional background information or references should be cited to provide more substantial explanations. In particular, short paragraphs in the Introduction, Methods, and Discussion sections should be naturally connected to improve coherence.
- In the Introduction, the phrase "Mañoso Hernando 20/11/24 7:48" appears to be an unrelated note or comment and should be removed.
- In the Methods section, the sentence "Interventionary studies involving animals or humans, and other studies that require ethical approval..." appears to be a general ethical guideline rather than content directly related to this study. Since it does not pertain to the research itself, it should be removed.
- The study is described as a pilot study in the Methods section. However, this is not explicitly stated in the title or abstract. To maintain consistency, it is necessary to clearly mention that this research is a pilot study in either the title or the abstract.
- The manuscript states that the CKCUEST was performed, but the rationale for analyzing EMG data is not adequately explained. Although it is mentioned that previous studies have used a similar method, further clarification is required to justify its appropriateness in the context of this study.
- The EMG signals were normalized as a percentage of the maximum voluntary isometric contraction (MVIC). However, detailed descriptions of electrode placement and signal processing procedures (e.g., filtering, RMS transformation, etc.) are lacking. This information is essential to ensure reproducibility and proper interpretation of the results.
- In section 2.4 (Statistical Analysis), only statistical methods related to mean comparisons are described. However, correlation analysis is mentioned in both the abstract and results section, despite no indication that such an analysis was conducted. This inconsistency should be addressed. If correlation analysis was not performed, it should be removed from the abstract and results. If it was conducted, it should be explicitly described in the Statistical Analysis section.
- Figure 1 is missing from the manuscript. The authors should add Figure 1, as its absence makes it difficult to understand the electrode placement. The supplementary file(s) do not contain Figure 1, so it must be included in the main text.
- While the modified CKCUEST may be suitable for patients with shoulder pain, the study only included healthy participants. This distinction should be explicitly stated to ensure that the study’s conclusions are not overstated. Additionally, since this is a pilot study, its limitations should be clearly acknowledged.
Author Response
Reviewer 2
The manuscript contains numerous short paragraphs consisting of only one or two sentences, which negatively impact readability. It is recommended to combine adjacent paragraphs that discuss similar topics to enhance the logical flow. If the content is insufficient, additional background information or references should be cited to provide more substantial explanations. In particular, short paragraphs in the Introduction, Methods, and Discussion sections should be naturally connected to improve coherence.
Response: We appreciate the reviewer’s valuable feedback on improving the manuscript's readability and coherence. In response, we have revised the Introduction, Methods, and Discussion sections by merging short paragraphs that addressed similar topics, ensuring a more natural flow of information. We believe these modifications have strengthened the manuscript’s structure and coherence and they have been highlighted in yellow in the manuscript.
In the Introduction, the phrase "Mañoso Hernando 20/11/24 7:48" appears to be an unrelated note or comment and should be removed.
Response: We appreciate the reviewer for identifying this issue. The phrase in question was an unintentional note that has now been removed from the manuscript. We apologize for any confusion this may have caused and thank you for your careful review.
In the Methods section, the sentence "Interventionary studies involving animals or humans, and other studies that require ethical approval..." appears to be a general ethical guideline rather than content directly related to this study. Since it does not pertain to the research itself, it should be removed.
Response: We appreciate the reviewer’s insightful observation. We acknowledge that the sentence in question represents a general ethical guideline rather than study-specific content. In response to this suggestion, we have removed the sentence to enhance the clarity and relevance of the Methods section. Thank you for your careful review and constructive feedback.
The study is described as a pilot study in the Methods section. However, this is not explicitly stated in the title or abstract. To maintain consistency, it is necessary to clearly mention that this research is a pilot study in either the title or the abstract.
Response: Thank you very much for your valuable correction. To ensure consistency throughout the manuscript, we have now explicitly mentioned that this research is a pilot study in the title, aligning it with the description provided in the Methods section. This modification enhances clarity and accurately reflects the scope of the study. Thank you for your insightful feedback. The modification has been highlighted in the Title Page as follows:
“Muscle Activation Differences Between CKCUEST and Modified CKCUEST: A Pilot Study.”
The manuscript states that the CKCUEST was performed, but the rationale for analyzing EMG data is not adequately explained. Although it is mentioned that previous studies have used a similar method, further clarification is required to justify its appropriateness in the context of this study.
Response: We appreciate the reviewer’s insightful comment. The rationale for analyzing EMG data during the CKCUEST is based on its ability to provide an objective assessment of muscle activation patterns, which are crucial for understanding neuromuscular demands during closed kinetic chain tasks. Previous studies have successfully employed EMG analysis in similar contexts to evaluate shoulder muscle recruitment and stabilization strategies. By incorporating EMG measurements, our study aims to extend this body of research by offering a deeper understanding of muscle activation differences between the CKCUEST and its modified version. It has been highlighted in yellow in the manuscript in line 67 as follows:
“The rationale for analyzing EMG data during the CKCUEST is based on its ability to provide an objective assessment of muscle activation patterns, which are crucial for understanding neuromuscular demands during closed kinetic chain tasks.”
The EMG signals were normalized as a percentage of the maximum voluntary isometric contraction (MVIC). However, detailed descriptions of electrode placement and signal processing procedures (e.g., filtering, RMS transformation, etc.) are lacking. This information is essential to ensure reproducibility and proper interpretation of the results.
Response: We sincerely appreciate the reviewer’s insightful comment. To enhance the reproducibility and clarity of our methodology, we have now included a detailed description of the electrode placement, highlighted in yellow in line 139, and signal processing procedures, highlighted in yellow in line 142, including filtering parameters and RMS transformation, in the Methods section. These additions ensure a more comprehensive understanding of our EMG normalization approach and facilitate proper interpretation of the results. Thank you for your valuable feedback.
“All electrodes were placed over the muscle bellies, in line with the orientation of muscle-fibers as previously described in the literature [33].”
“For the signal processing, it was amplified (gain=1,000), rectified and then band-pass filtered (Butterworth 20-400 Hz). The envelope of the whole EMGs signal was calculated with a fixed (for the MVIC test) and dynamic (for the CKCUEST test) 500 ms. time-window. Finally, the normalised root mean square (%RMS) of the 5 central seconds of each CKCUEST stages was calculated and expressed as a percentage (%) of the corresponding MVIC test (%RMS).”
In section 2.4 (Statistical Analysis), only statistical methods related to mean comparisons are described. However, correlation analysis is mentioned in both the abstract and results section, despite no indication that such an analysis was conducted. This inconsistency should be addressed. If correlation analysis was not performed, it should be removed from the abstract and results. If it was conducted, it should be explicitly described in the Statistical Analysis section.
Response: We would like to thank the reviewer for the valuable comment and apologise for the mistake. Indeed, there is no correlation analysis but a comparison analysis and therefore, we have modified in the abstract, results section and tables the error. It is highlighted in yellow in the abstract, results section, line 228, and Table 2 and 3.
“Comparison between laterality and the percentage of activation of the infraspinatus, anterior deltoid and upper trapezius muscles in the CKCUEST test are shown in Table 3.”
Figure 1 is missing from the manuscript. The authors should add Figure 1, as its absence makes it difficult to understand the electrode placement. The supplementary file(s) do not contain Figure 1, so it must be included in the main text.
Response: We appreciate the reviewer’s careful assessment. Figure 1 was unintentionally omitted from the manuscript. We have now included it in the main text to ensure clarity, particularly regarding electrode placement. Thank you for bringing this to our attention.
While the modified CKCUEST may be suitable for patients with shoulder pain, the study only included healthy participants. This distinction should be explicitly stated to ensure that the study’s conclusions are not overstated. Additionally, since this is a pilot study, its limitations should be clearly acknowledged.
Response: We appreciate the reviewer’s insightful comment. We acknowledge that our study was conducted exclusively on healthy participants, and we have now explicitly stated this limitation to ensure that our conclusions are not overstated regarding its applicability to patients with shoulder pain. Additionally, as this is a pilot study, we have further elaborated on its limitations in the Discussion section, emphasizing the need for future research in clinical populations. It has been added to the manuscript and highlighted in yellow in line 305. Thank you for your valuable feedback, which has helped strengthen the clarity and validity of our findings
“At present, this research is a pilot study, where the sample size calculated for the main variable has not been reached. Furthermore, while the modified CKCUEST may be suitable for patients with shoulder pain, the study only included healthy participants. For these reasons, the results of this research should be taken with caution, reducing the external validity of the data and therefore its interpretation.”
We appreciate the reviewer's feedback and are open to further clarifications or additional analyses to strengthen the study.
Reviewer 3 Report
Comments and Suggestions for Authors
Dear Author;
I have some questions about your study:
You mention that the study was a pilot study with healthy subjects, so the title should be changed to "Muscle Activation Differences Between CKCUEST and Modified CKCUEST: A pilot study".
The mean age of the participants should be added to the abstract.
Page 1 line 44 "Mañoso Hernando 20/11/24 7:48" This should be deleted.
"For this pilot study, a total of 20 healthy subjects, 10 male and 10 female participants" How did you determine the sample size? Did you analyse the post hoc power?
You mention that "a total of 20 healthy subjects were included". However, your exclusion criteria include being over 65 years of age and having dementia....etc.
The inclusion criteria have been revised to be between 18-65 years of age (I think 65 years is too old for a healthy subject).
Why did you use the IPAQ?
Page 5 line 203-204 there is no ethical information.
There is no gender difference in the results section or tables for EMG activity.
What are the strengths of this study?
What protocol do you think should be used?
Please check the references (example: 11-43)
Author Response
Reviewer 3:
Dear Author;
I have some questions about your study:
You mention that the study was a pilot study with healthy subjects, so the title should be changed to "Muscle Activation Differences Between CKCUEST and Modified CKCUEST: A pilot study".
Response: Thank you very much for your valuable correction. To ensure consistency throughout the manuscript, we have now explicitly mentioned that this research is a pilot study in the title, aligning it with the description provided in the Methods section. This modification enhances clarity and accurately reflects the scope of the study. Thank you for your insightful feedback. The modification has been highlighted in the Title Page as follows:
“Muscle Activation Differences Between CKCUEST and Modified CKCUEST: A Pilot Study.”
The mean age of the participants should be added to the abstract.
Response: We appreciate the reviewer’s suggestion. In response, we have now included the mean age of the participants in the abstract to provide a more complete characterization of the study population, it has been highlighted in yellow in the abstract section as follows. Thank you for your valuable feedback.
“Ten male (age: 26.6 and 10 female participants (age: 24.2 were recruited from a University setting.”
Page 1 line 44 "Mañoso Hernando 20/11/24 7:48" This should be deleted.
Response: We appreciate the reviewer for identifying this issue. The phrase in question was an unintentional note that has now been removed from the manuscript. We apologize for any confusion this may have caused and thank you for your careful review.
"For this pilot study, a total of 20 healthy subjects, 10 male and 10 female participants" How did you determine the sample size? Did you analyse the post hoc power?:
Response: We appreciate the reviewer’s important question regarding sample size determination. As this study was designed as a pilot study, the sample size was chosen based on previous similar research and methodological feasibility rather than a formal power calculation. The required sample size was estimated based on previous recommendations for pilot studies (Hertzog MA,;2008) and it has been highlighted in yellow in the manuscript in line 82 as follows. Thank you for your valuable feedback.
“The required sample size was estimated based on previous recommendations for pilot studies (28)”
-Hertzog MA. Considerations in determining sample size for pilot studies. Res Nurs Health. 2008 Apr;31(2):180-91. doi: 10.1002/nur.20247. PMID: 18183564.)
You mention that "a total of 20 healthy subjects were included". However, your exclusion criteria include being over 65 years of age and having dementia....etc.
Response: Thank you for your observation. The exclusion criteria, including age over 65 years and conditions such as dementia, were established to ensure a homogeneous and truly healthy sample, minimizing potential confounding factors. These criteria align with our previous study (Manoso-Hernando D, Bailón-Cerezo J, Elizagaray-García I, Achútegui-García-Matres P, Suárez-Díez G, Gil-Martínez A. Cervical and Thoracic Spine Mobility in Rotator Cuff Related Shoulder Pain: A Comparative Analysis with Asymptomatic Controls. J Funct Morphol Kinesiol. 2024 Jul 24;9(3):128. doi: 10.3390/jfmk9030128. PMID: 39189213; PMCID: PMC11348207.) where we used the same methodological approach to maintain consistency and comparability of results.
Why did you use the IPAQ
Response: We appreciate the reviewer’s inquiry regarding the use of the International Physical Activity Questionnaire (IPAQ). The IPAQ was chosen as a standardized and validated tool to assess participants' physical activity levels, ensuring that groups were comparable in terms of habitual activity. This was particularly important to minimize potential confounding effects of physical activity on muscle activation patterns. By using the IPAQ, we aimed to enhance the internal validity of our study and ensure that differences observed in muscle activation were not influenced by variations in baseline physical activity levels. We have now clarified this rationale in the manuscript in line 241 and it has been highlighted in yellow as follows:
“Equally, no statistical differences on IPAQ scores were found between men and women, given that physical activity can influence neuromuscular responses, using the IPAQ allowed us to categorize and control for activity levels, ensuring that any differences observed in muscle activation were due to the studied variables rather than variations in habitual physical activity [41]”.
Page 5 line 203-204 there is no ethical information.
Response: We appreciate the reviewer’s insightful observation. We acknowledge that the sentence in question represents a general ethical guideline rather than study-specific content. In response to this suggestion, we have removed the sentence to enhance the clarity and relevance of the Methods section. Thank you for your careful review and constructive feedback.
There is no gender difference in the results section or tables for EMG activity. Response: We appreciate the reviewer’s insightful observation. In response, we have now included a table presenting the EMG activity data stratified by sex to provide greater transparency regarding potential gender differences. This addition enhances the comprehensiveness of our results and allows for further interpretation of muscle activation patterns. It is included as a Table 4 and in the manuscript highlighted in yellow in line 232. Thank you for your valuable feedback.
“Comparison between sex and the percentage of activation of the infraspinatus, anterior deltoid and upper trapezius muscles in the CKCUEST test are shown in Table 4.
(Table 4 near here)”
What are the strengths of this study?
Response: Thank you very much for letting us clarifies this point. We have added a paragraph shedding light on the article´s strength. It is highlighted in yellow in the manuscript in line 290 as follows:
“Our findings provide robust evidence that the modified CKCUEST induces significantly lower EMG activity in the infraspinatus, anterior deltoid, and upper trapezius muscles compared to the CKCUEST. This suggests that the modified CKCUEST is a highly suitable option for physical functional assessment in individuals with shoulder pain or low physical activity levels. Furthermore, a structured progression from the modified CKCUEST to the CKCUEST should be considered in shoulder rehabilitation to optimize tissue loading capacities, as this test has demonstrated a strong correlation with shoulder strength [44].”
What protocol do you think should be used?
Response: We appreciate the reviewer’s insightful comment regarding the protocol selection. Based on our findings, we believe that the modified CKCUEST should be considered a suitable option for physical functional assessment, particularly in individuals with shoulder pain or lower physical activity levels. Given its ability to induce lower EMG activity in key shoulder muscles, this protocol may be preferable in clinical and rehabilitative settings where reducing excessive muscle activation is desired. However, a structured progression from the modified CKCUEST to the standard CKCUEST could be beneficial in rehabilitation to optimize tissue loading capacities, as this test has been shown to correlate with shoulder strength. We have clarified this rationale in the manuscript, in line 293, to provide a more comprehensive explanation of our approach. Thank you for your valuable feedback
“This suggests that the modified CKCUEST is a highly suitable option for physical functional assessment in individuals with shoulder pain or low physical activity levels. Furthermore, a structured progression from the modified CKCUEST to the CKCUEST should be considered in shoulder rehabilitation to optimize tissue loading capacities, as this test has demonstrated a strong correlation with shoulder strength [44].”
Please check the references (example: 11-43)
Response: Thank you for pointing this out. In response, we have made the necessary corrections in the manuscript to address this point. Thank you for your valuable input.
We appreciate the reviewer's feedback and are open to further clarifications or additional analyses to strengthen the study.
Round 2
Reviewer 2 Report
Comments and Suggestions for Authors
The authors have provided sincere and thoughtful responses to all the comments raised during the previous round of review. They have also made appropriate revisions to the manuscript in accordance with the reviewers' suggestions. The changes have improved the clarity, structure, and overall quality of the manuscript. After carefully reviewing the revised version, I find that the authors have adequately addressed the concerns raised in the initial review. Therefore, I support the acceptance of this revised manuscript for publication.